

# cloudbandPy v1.0: an automated algorithm for the detection of tropical-extratropical cloud bands

Romain Pilon[1] and Daniela I.V. Domeisen[1,2]

[1]Institute of Earth Surface Dynamics, University of Lausanne, Lausanne, Switzerland
[2]Institute for Atmospheric and Climate Science, ETH Zürich, Zürich, Switzerland

**Correspondence:** romain.pilon@unil.ch

**Abstract.** Persistent and organized convective cloud systems that arise in convergence zones can lead to the formation of synoptic cloud bands extending from the tropics to the extratropics. These cloud bands are responsible for heavy precipitation and are often a combination of tropical intrusions of extratropical Rossby waves and of processes originating from the tropics. Detecting these cloud bands presents a valuable opportunity to enhance our understanding of the variability of these systems and the underlying processes that govern their behavior and that connect the tropics and the extratropics. This paper presents a new atmospheric cloud band detection method based on outgoing longwave radiation using computer vision techniques, which offers enhanced capabilities to identify long cloud bands across diverse gridded datasets and variables. The method is specifically designed to detect extended tropical-extratropical convective cloud bands, ensuring accurate identification and analysis of these dynamic atmospheric features in convergence zones. The code allows for easy configuration and adaptation of the algorithm to meet specific research needs. The method handles cloud band merging and splitting, which allows for an understanding of the life cycle of cloud bands and their climatology. This algorithm lays the groundwork for improving our understanding of the large-scale processes that are involved in the formation and life cycle of cloud bands and the connections between tropical and extratropical regions, and to evaluate the differences in cloud band types between different ocean basins.

## 1 Introduction

Convergence zones can be described as broad bands of persistent, quasi-stationary convectively active clouds, which are aligned from northwest to southeast and from southwest to northeast, in the southern and northern hemispheres, respectively. They typically occur throughout the respective wet season. The water and energy cycle within these convergence zones is strongly modulated by individual storm events that are embedded in the synoptic circulation, similar to other convective regions (Rickenbach and Rutledge, 1998; Schumacher and Houze, 2003; Roca et al., 2010, 2014). These individual storm events play a crucial role in shaping the distribution and intensity of precipitation, affecting the overall hydrological and atmospheric dynamics within the convergence zone. There are four tropical convergence zones, which are the Baiu frontal zone (BFZ), which is here considered jointly with the Meiyu frontal zone over East Asia, (Kodama, 1992, 1993; Li et al., 2018), the South Atlantic convergence zone (SACZ) (Carvalho et al., 2004; Villela, 2017), the South Indian convergence zone (SICZ), (Cook, 2000) and the South Pacific Convergence Zone (SPCZ), which is the most vast and intense (Vincent, 1994; Matthews, 2012; Brown



et al., 2020). The convergence zones are formed by low-level wind convergence, which causes an accumulation of moist air and eventually leads to an upward movement of air, convective activity and intense rainfall (Streten, 1973). As convective storms progress within the convergence zone, they intensify moisture convergence, fostering heightened cloud formation and precipitation (Hudson, 1971; Houze, 1997; Tsuji et al., 2021). The synoptic circulation provides the large-scale framework for these individual storm events and influences the spatial organization and evolution of the storms within the convergence zone.

The SPCZ is the primary region where persistent deep tropical convection frequently merges with troughs from the mid-latitude circulation. Its largest extent is observed during austral summer (from October to April) (Vincent, 1994). Convergence zones allow for the organization of mesoscale convective systems (MCSs) (Matthews et al., 1996; Takahashi and Battisti, 2007; Oueslati and Bellon, 2013), which consist of an ensemble of cumulonimbus towers and anvils that become organized on a scale larger than the individual convective core (Houze Jr., 2004), and that form long, narrow bands that extend from the tropics to

the subtropics with a northwest-southeast orientation in the southern hemisphere and a southwest-northeast orientation in the northern hemisphere (sometimes referred to as a diagonal tilt). Given that the cloud bands over the SPCZ are a significant contributor to precipitation, especially on Pacific islands, it is crucial to document them in order to gain a better understanding of the water and energy cycle.

Atmospheric feature extraction and tracking can aid in this effort to identify cloud bands by creating a reduced dataset

that allows for more efficient visualization and statistical evaluation of cloud bands. Originating from image processing and computer vision Zucker (1976), this technique has proven to be a valuable tool for identifying and analyzing features of interest in various fields. In the extratropics, atmospheric feature detection has been successfully applied to study mid-latitude cyclones (Ulbrich et al., 2009), jet stream features (Limbach et al., 2012), and extreme precipitation events associated with potential vorticity streamers and integrated water vapor transport structures (de Vries et al., 2018). (Post et al., 2003) and (Limbach et al.,

2012) provide comprehensive overviews of conceptual views on feature identification and tracking in atmospheric sciences.

In addition to their application in the extratropics, image processing techniques have also been instrumental in the detection of organized convective cloud systems. These cloud systems can be identified using satellite data at a global scale, using measures such as infrared brightness temperature (Fiolleau and Roca, 2013; Roca et al., 2014; Huang et al., 2018; Laing and Michael Fritsch, 1997; Williams and Houze, 1987) or radar reflectivity (Nesbitt et al., 2006; Kummerow et al., 2011; Houze Jr.

et al., 2015), by combining brightness temperature and precipitation from the Global Precipitation Measurement dataset (Feng et al., 2021, 2022). These studies focused on the life cycle of MCSs.

Another proxy of deep convection is the outgoing long-wave radiation (OLR). Deep convective clouds are usually identified by their cold cloud tops, which emit low OLR values. These clouds possess a dense and vertically extended structure, leading to reduced emission of longwave radiation. Consequently, low OLR values often signify the presence of high, thick clouds,

including deep convective clouds. OLR has been used to study organization of convection (Holloway and Woolnough, 2016), the energy balance and tropical convection (Hartmann et al., 2001), to estimate deep convection (Waliser et al., 1993; Zhang et al., 2017), or to study the link between convective clouds, anvils and cirrus clouds (Massie et al., 2002; Sokol and Hartmann, 2020).



Larger cloud systems, such as tropical-extratropical cloud bands, are often detected using satellite imagery or reanalysis data using OLR thresholding, i.e. by dividing OLR into two groups with low and high values, respectively, to distinguish different clouds according to their radiation characteristics Kodama (1992, 1993). Low OLR values typically indicate the presence of convective clouds, while high OLR values are associated with regions where the atmosphere is relatively clear of clouds or where the cloud cover consists of thin or low-level clouds. More recent work has focused on the detection of cloud bands over the SACZ (Zilli and Hart, 2021; Rosa et al., 2020) and over the SICZ (Hart et al., 2012, 2018).

Despite the different available detection methods of tropical-extratropical cloud bands, all studies focus on the subtropical part of the cloud bands. Most of the available open-source tools are tailored for the subtropics, and do not treat merging and splitting of cloud bands, are not optimized to work with different data, and thresholds are dependent on the specific dataset.

This study aims to overcome these shortcomings by introducing a user-friendly Python software package designed to detect and track the life cycle of extended tropical-extratropical cloud bands. Specifically, the software package tracks the inheritance of cloud bands, allowing them to inherit properties from previous cloud bands over time. This time dependency allows for a representation of a cloud band's life cycle. The algorithm is applicable to various types of gridded data and can operate at different temporal and spatial scales. It explicitly handles the merging and splitting of features and includes visualization and analysis of detected cloud bands. It also includes visualization and analysis tools for the detected cloud bands. The goal of this paper is to describe this algorithm and to demonstrate its capabilities with examples applied to reanalysis data, and to identify the limitations of thresholding image-segmentation methods to study these large-scale cloud bands.

## 2 Data and methods

### 2.1 Dataset

We use the ERA5 global reanalysis dataset (Hersbach et al., 2020) from the European Centre for Medium-Range Weather Forecasts (ECMWF) as input data for our cloud band detection algorithm. It provides hourly estimates of a large number of atmospheric, land and oceanic climate variables on a 30km grid and at 137 vertical levels. In this study, cloud bands are detected from 1959 to 2021 using ERA5 OLR data at 3-hour intervals regridded to a regular latitude-longitude grid of 0.5 degrees at a global scale.

### 2.2 Cloud band identification

Low OLR values generally correspond to the cold cloud shields of convective systems, which include the core of the convective system and its anvil. In this study, the algorithm uses OLR data to identify tropical-extratropical diagonal cloud bands associated with deep convective cloud systems. The algorithm includes a threshold-based segmentation and utilizes a morphological approach, which modifies the shape of objects in an image and extract valuable information, such as their geometric characteristics. Segmentation refers to dividing the image into regions of pixels based on the OLR threshold and has been used in other studies such as in SACZ, SICZ automated detection algorithms (Hart et al., 2012, 2018; Rosa et al., 2020). Our method





of tracking the time dependency of OLR associated with cloud bands takes into account the full life cycle of cloud bands. In addition, applying the method globally and across a wide range of latitudes (from the tropics to the extratropics) allows us to extend the cloud band climatology to a global scale, enabling a comparison between different basins and also to highlight the limitations of the here presented tool.

We apply the basic workflow of image data analysis on OLR data, which are: smoothing, binarization of the image, setting
a threshold, applying morphological operations and labeling. Version 3 of Python is used for all steps. Figure 1 shows the step-by-step process of the cloud band identification developed in this study, which is hereafter explained in detail.

First, we calculate the daily mean of the 3-hourly OLR data (Figure 1a). This average is used as the smoothing procedure that prevents over-segmentation of cloud systems, i.e. temporal smoothing increases the connectivity between low OLR regions. Such over-segmentation usually occurs at the early stage of cloud band formation and at its last stage before splitting (see
section 2.4) or disappearance, when convective systems start to organize. Instead of using the temporal smoothing employed here, a spatial box smoothing may be needed when using different periods and may have to be adjusted for data sets of different temporal and/or horizontal resolution.

Secondly, a threshold is used to differentiate regions of low and high OLR values through binarization of the data (Figure 1a,b). Choosing an appropriate value for thresholding images can be challenging. The main problem with thresholding is that
it considers only the intensity for each single pixel, but not any relationships between the pixels. Following Massie et al. (2002) on the distribution of tropical cirrus clouds associated with deep convective anvils, contiguous areas with smoothed OLR below values of 210 W.m$^2$ are here identified as distinct cloud systems (Figure 1a). The limitations of using thresholding methods are described in the discussion section.

The above threshold is chosen for two reasons: The first reason is that the cloud band systems we want to identify are
mainly convective and areas of frequent convection are often accompanied by a high occurrence of cirrus clouds (Sassen et al., 2009; Schoeberl et al., 2019; Nugent et al., 2022), which have OLR values below this threshold. Moreover we want to take into consideration anvils from deep convective clouds. The second reason is that we aim to prevent convective cloud systems, extending from the tropics to the extratropics, from forming a single region of low OLR values with upper tropospheric cold clouds from mid-latitudes and polar regions that extend over large areas.

Next, each cloud system undergoes a morphological dilation (Figure 1c), adding pixels to the boundaries of each object in an image, expanding them and increasing their size. Thereby, we assume that surrounding warmer cloud pixels (higher OLR values) are preferentially grouped to the larger cloud systems and can be associated with convective systems (e.g. anvils) from a cloud band. Region growing allows us to merge clouds that occupy neighboring pixels and can prevent the subsequent labeling step from separating clouds that could be part of a single cloud band.

As a next step, all cloud systems are labeled using a 8-connected components labeling method (Figure 1d). Connectivity refers to the relationship between pixels and their neighbors, which can form objects or groups. Here, a pixel is considered to be connected to its eight neighbors. The labeling process involves iterating over all pixels in the image and assigning labels to connected objects. The labeled systems are treated as image-like arrays allowing for measuring properties that are not dependent on the spatial resolution or the projection of the input data. To speed up the last step of the algorithm, smaller cloud





systems are filtered out (Figure 1e), where cloud systems below the threshold of $10^5$ km$^2$ are subsequently excluded. This threshold value is comparable to that of an organized convective system, which typically has an area of $10^4$ km$^2$ or larger (Roca and Ramanathan, 2000; Houze Jr., 2004). To ensure that cloud systems are not excessively removed from the labelled systems, we visually examine a wide selection of time steps and values to determine a suitable area threshold.

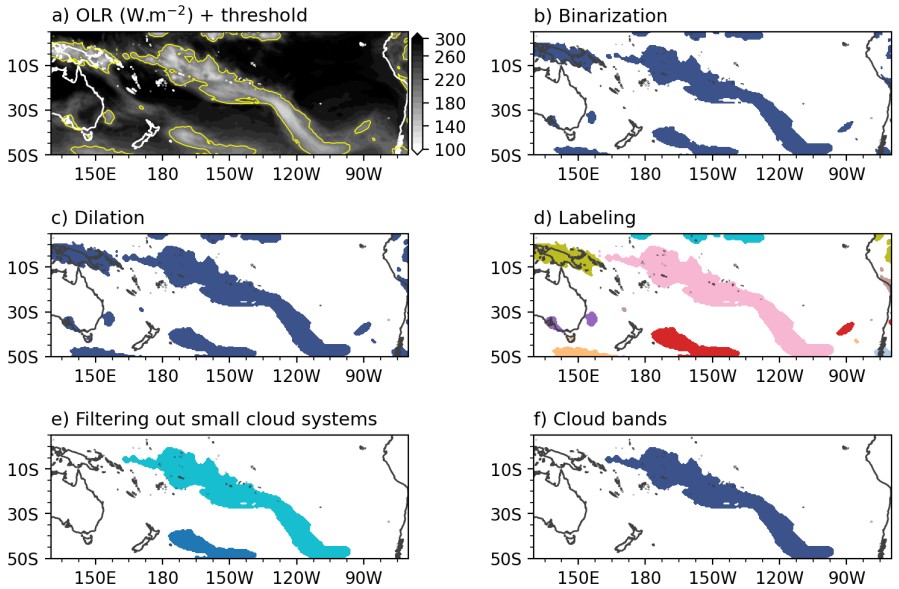

**Figure 1.** Case study example of the cloud band identification processing steps for 18 April 2016 using OLR. a) Daily mean OLR data in shading and 210 W.m$^2$ isocontour in yellow, b) binarization, c) morphological dilation, d) connected component labeling with different colors indicating distinct identified features, e) cloud systems with a size below a threshold value are removed, f) cloud band after filtering.

Finally, each identified feature is transformed into an object. For each object, some properties can be calculated by drawing
an ellipse around it, such as its orientation (that is, the tilt of a cloud band with respect to the latitude / longitude grid) measured from the ellipse's centroid location and the orientation of its axes. Specifically, in this study, we define a tropical-extratropical diagonal cloud band as a cloud system with the following properties:

(1) Its major axis has to exhibit an orientation between -5 and -90° in the southern and between 5 and 90° in the northern hemisphere. Cloud bands without a strong enough latitudinal tilt, i.e. with an orientation along a latitude circle, are
filtered out to prevent labeling cloud systems belonging for example to the Inter-Tropical Convergence Zone (ITCZ) as a diagonal cloud band.

(2) Each cloud band must cross 23.5° North or South, which defines the extent of the tropics. By doing so, we assume that cloud bands must have a minimal extent and must cover tropical and extratropical regions. Furthermore, for cloud bands to have a minimal extent, their northernmost and southernmost latitudes have to lie equatorward of 20°S and poleward





of 27°S, respectively, in the Southern Hemisphere, and equatorward of 20°N and poleward of 27°N, respectively, in the
     Northern Hemisphere.

## 2.3  Domains of detection and limitations of a threshold-based detection

Although the detection workflow is here initially designed for the South Pacific Ocean and tested in both hemispheres, it is
advisable to establish a specific domain for each basin based on the prevailing processes. This is particularly relevant for the
detection of cloud bands over convergence zones, for which the workflow is specifically developed. Since the method is based
on an OLR threshold, all regions covered by cold clouds can influence the identification and lead to a false identification of
cloud bands by merging actual cloud bands with other cloud cover types (from the mid-latitudes, or from the ITCZ), despite
the orientation criterion.

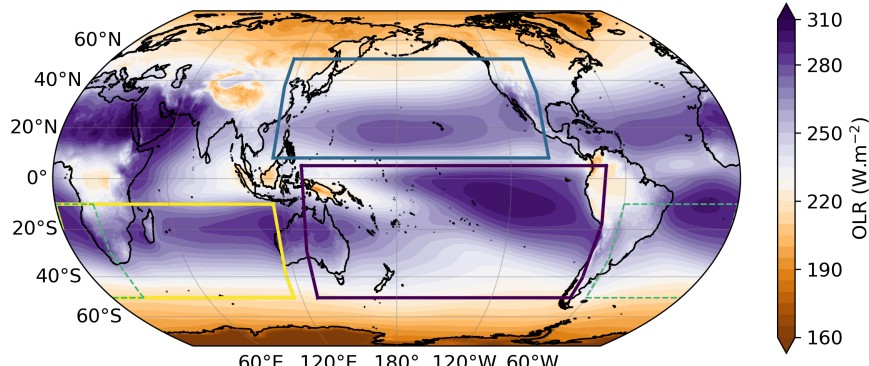

**Figure 2.** Average OLR (in W/m$^2$) from 1959 to 2021 from ERA5 data. The rectangles correspond the domains used in this study.

Moreover, low OLR values of cold clouds located over high altitude terrains such as the Tibetan plateau (Su et al., 2000)
may connect with other low OLR regions emerging from the South Asian monsoon (Figure 3a) and from the BFZ. Another
example of erroneous detection is the presence of low OLR values above the Andes and the Bolivian high, which can connect
clouds systems from the ITCZ with clouds from the SACZ (Lenters and Cook, 1997; Villela, 2017) (blue contour in Figure
3b). Additionally, since the SPCZ may be seen as an extension of the ITCZ, the algorithm may identify the elongated region
of low OLR values as a cloud band, influenced by the cloud orientation above the SPCZ (yellow contour in Figure 3b). In such
cases, the presence of diagonal cloud bands may be an artifact of the algorithm. Therefore, it is crucial to exercise caution when
interpreting cloud band detection over these regions.

A guideline would be to define specific domains based on a climatology of OLR, as shown in Figure 2. In this figure,
we suggest four domains that encompass the four convergences zones. It is worth noting that the North Atlantic basin is not
included in the selection as it does not exhibit a distinct convergence zone. The domains will be subsequently referred to
by the names of the respective ocean basins. The North Pacific domain ([115°E/100°W, 50°N/8°S]) covers the eastern part
of the North American continent and offers the potential to identify cloud formations related to atmospheric rivers (Neiman



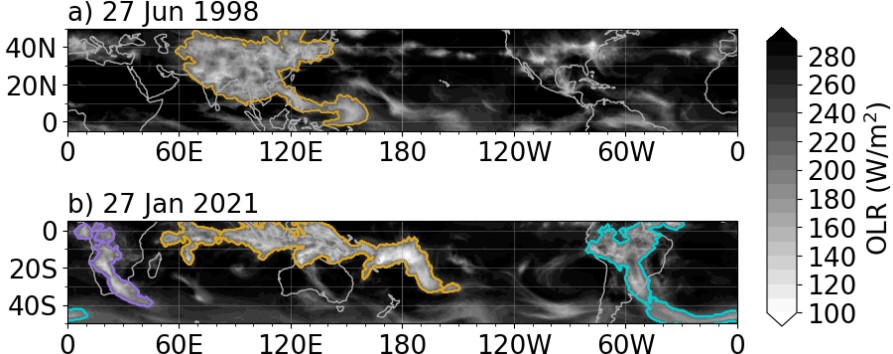

**Figure 3.** Illustration of problematic cloud band detection on (a) 27 June 1998 and (b) 27 January 2021, highlighting the significance of setting domains based on convergence zones. Shading corresponds to OLR data and colored contours represent detected cloud bands.

et al., 2008; Guan et al., 2023) that extend to the western part of the North American continent. The South Pacific domain ([130°E/70°W, 5°N/50°S]) encompasses the SPCZ as well as potential cloud bands extending towards the South American continent. One possible shortcoming of using the algorithm in this region is the identification of tropical cyclones within cloud

bands and the potential merging of cloud cover between the westernmost part of the ITCZ in this domain with cloud bands over the SPCZ. The South Atlantic domain ([60°W/20°E, 10°S/50°S]) primarily covers the SACZ. To avoid potential merging of the Bolivian anticyclone and the ITCZ with cloud bands over the SACZ, the northwestern portion of the convergence zone is intentionally excluded. The South Indian Ocean domain ([0°/115°E, 10°S/50°S]) encompasses the SICZ and allows the detection of potential elongated cloud bands.

To enable the detection of tropical-extratropical cloud bands globally, the latitudinal range must be adjusted and set from 10° to 50°N and S, taking into account the varying location of the ITCZ over time, which can occasionally extend up to 20°N/S (Cook, 2000; Waliser and Jiang, 2015; Liu et al., 2020). However, in this case, the algorithm should not flag features resembling the ITCZ that are located poleward of 20° as cloud bands (second criterion of the last step of the detection in section 2.2). Limitations of global detection are discussed further in section 3.2.

**2.4 Inheritance tracking**

Here our tracking method of cloud bands over time uses a simple area overlap method. Two-dimensional objects can be linked together across adjacent time frames based on the amount of overlap between them. Cloud bands with an area overlap of more than 40% between two consecutive time steps are considered as the same cloud band. The value of 40% is chosen because of the quasi-stationary nature of cloud bands. The overlap area is calculated in both time directions (i.e., from time $t-1$ to

time $t$ and from $t$ to $t+1$), allowing us to determine the history and future of each cloud band and if a cloud band is either growing or shrinking in size. That is, the merging or splitting of cloud bands are treated explicitly. An example of splitting and merging of cloud bands is shown in Figure 4. In this figure, cloud bands are detected from 20 to 28 February 2021 over the



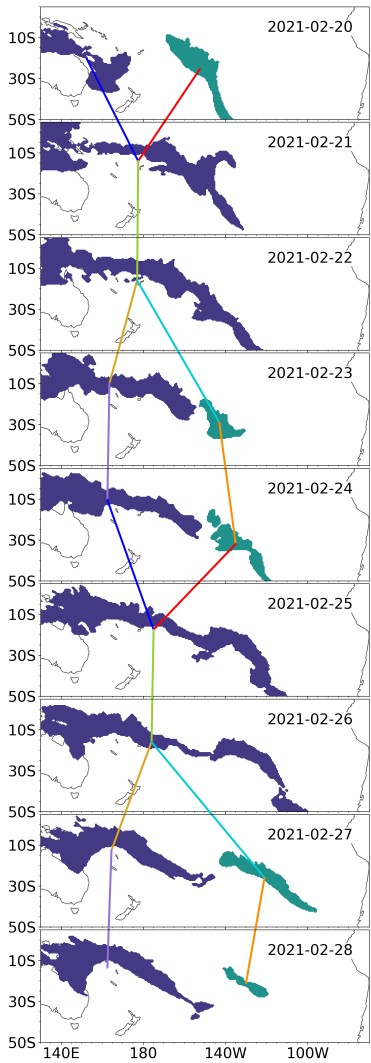

**Figure 4.** Example of splitting and merging of cloud bands, and their inheritance tracking from 20 to 28 February 2021 in the South Pacific Ocean. Lines connect the cloud bands' centroid locations. Colors are chosen to visualize merged (blue) versus split (blue versus green) cloud bands.

South Pacific. Initially, a cloud band in the West Pacific merges with another cloud band originating from the central Pacific between February 20th and 21st. Subsequently, the resulting cloud band persists for two days before splitting into two separate
cloud bands on February 23rd and 24th. These two bands then merge again on February 24th and 25th. It becomes apparent that the eastern segment of the cloud band slowly separates on February 25th and 26th, ultimately appearing as two distinct cloud bands on the 27th. By February 28th, the easternmost cloud band begins to diminish in size, and the following day it is no longer detected (not shown in the figure).





Tracking methods in general provide information about the lifetime of the tracked features (Bengston et al., 1995; Camargo
and Zebiak, 2002; Sugi et al., 2002; Chauvin et al., 2006). In this study, tracking tropical-extratropical cloud bands means
knowing their inheritance from one time step to another. The tracking here does not provide information about the lifetime
of a specific cloud band. Therefore, it does not indicate how long a specific cloud band lasts. This concept can therefore be
inadequate, since cloud bands are quasi-permanent and quasi-stationary, particularly when considering cloud bands over the
SPCZ (Streten, 1973; Vincent, 1994; Brown et al., 2020). Focusing on the lifetime of a specific cloud band could lead for the
workflow to identify long periods where a single cloud band persisted, despite the likelihood that the initial identified cloud
band may have split into two different ones and merged with another one, like for example with a trough intruding from the
mid-latitudes (Kiladis et al., 1989).

## 2.5  Implementation and Configuration

The method developed in this study is embedded in a Python package called *cloudbandPy*. Installation can be done using
the Pip package management system (The Pip Development Team, 2021) or using the Conda environment setup (Anaconda,
Inc., 2016). Environment, requirements files and installation instructions are provided here: https://github.com/romainpilon/
cloudbandPy. All user-defined parameters are stored in a YAML file (https://yaml.org/) for easy access and modification. The
configuration file specifies dates, study domains, data location and which steps to run. As an example, loading one year of 0.5°
horizontal resolution OLR data from ERA5, the detection of cloud bands and writing files that contain all detected cloud bands
takes less than one minute using an AMD EPYC processor.

## 3  Application to ERA5 reanalysis data

We envisage that the here presented algorithm will be used for assessing tropical-extratropical connections in different datasets
including global climate simulations, and we now present a few examples of such applications. All examples briefly discussed
in this section are included in the public repository linked to in the *Code and data availability* statement.

## 3.1  Case study

Among the cloud bands in the literature that satisfy the here chosen criteria, we select a well-documented cloud band resulting
from tropical-extratropical interaction, which was extensively investigated by Knippertz (2005) (hereafter referred to as KP05).
This cloud band ("tropical plume") is described as an elongated band of upper- and mid-level cloud formation stretching from
the central tropical Atlantic ocean to the northern African continent, and was accompanied by a subtropical jet streak, from 29
March to 1 April 2002. The evolution of this specific cloud band with a latitudinal tilt is shown with daily average of OLR in
Figure 5 (a-d) and in satellite infrared imagery (Figure 2 from Knippertz (2005)).

Although this basin falls outside of our primary areas of interest, which focus on basins with a tropical-extratropical con-
vergence zone, this cloud band serves as an excellent test bed, and provides insights into the behavior of our algorithm. For
this particular event, we perform the detection process in the entire northern hemisphere. In Figure 5 (e-h), both cloud bands



and the identified features (obtained from the step illustrated in Figure 1e) are displayed. On March 29, 2005, our algorithm
detects a cloud band in the western Atlantic with a strong inclination, spanning from the Caribbean to the western coasts of
Canada. The following day, this cloud band shifts poleward and is no longer classified and labeled as a tropical-extratropical
cloud band by our algorithm (its southern tip is at 21°N on March 30 and at 23°N on March 31); the northern section of this
unlabeled cloud band shifts eastward along with the mid-latitude circulation. The mid-latitude segment of the unlabeled cloud
band dissipates by March 31st, and on April 1st, no further features are detected.

In KP05, the cloud band that reaches western Africa becomes visually apparent on March 30, 2005. Our detection identifies
a feature initially in the central tropical Atlantic ocean, but it is disregarded due to insufficient spatial extent (based on the
latitudinal extent criterion). However, the cloud band itself is successfully detected on March 31.

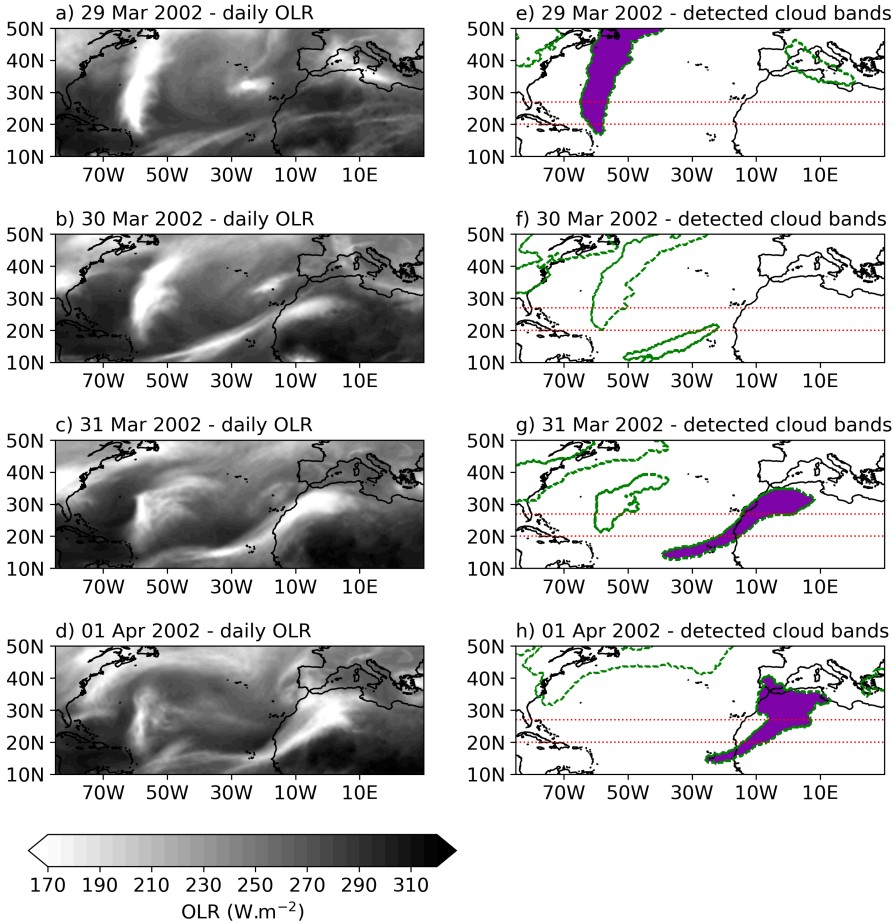

**Figure 5.** Snapshots of (a-d) OLR and of (e-h) associated detected cloud bands (shading) and of identified features, i.e. potential cloud bands
(dashed contours) over the north Atlantic basin from 29 March to 1 April 2002. The horizontal lines represent the latitudinal lines that cloud
bands must cross to be defined as a tropical-extratropical cloud band.



## 3.2 Spatial distribution of cloud bands

Long-term global detection allows for an illustration of the spatial distribution of tropical-extratropical cloud bands. The spatial distribution of cloud bands is illustrated in Figure 6 showing a climatology of the number of cloud band days per year per grid point from 1959 to 2021). The figure clearly illustrates the presence of the four tropical convergence zones, as indicated by the highest number of cloud band days per year. The SPCZ and the SICZ exhibit maxima of 49 and 28 cloud band days per year, respectively. Cloud bands from the BFZ merge, here, with cold cloud cover from the Tibetan plateau and from the South
Asian monsoon, which is also labeled as cloud bands. In the northeast Pacific region, from Hawaii to south of California, notable occurrences of cloud bands, commonly referred to as atmospheric rivers, are observed (on average from one to two weeks per year). It is worth noting that long cloud spirals trailing away from the centers of tropical cyclones may also be identified as cloud bands. Additionally, in the eastern tropical Atlantic Ocean and the western Sahara, there are instances of cloud bands, including the one discussed in KP05 and depicted in the previous section. The North Atlantic basin does not
exhibit a convergence zone, hence it contains only few cloud bands. The northern Caribbean experiences approximately 14 cloud band days per year, potentially influenced by tropical cyclones, as compared to 21 cloud band days in the Northeast Pacific region, which is also influenced by tropical cyclones, and by atmospheric rivers.

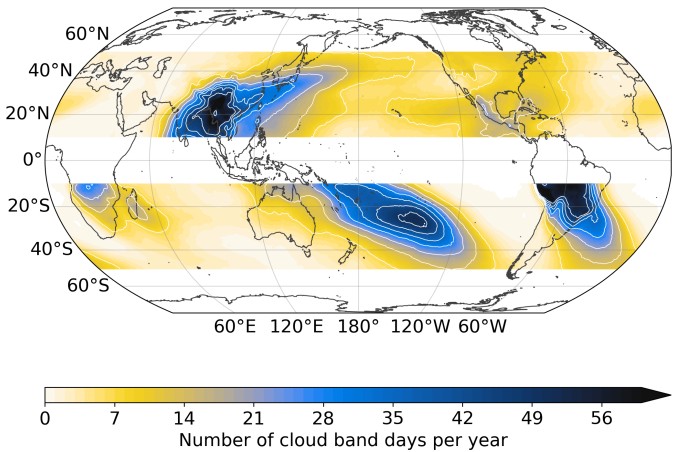

**Figure 6.** Number of cloud band days per year averaged from 1959 to 2021. Contour interval: 7 cloud band days per year.

To mitigate the issue of false detections (i.e., non-diagonal tropical-extratropical cloud bands) in spatial distributions, it is preferable to conduct the detection process separately for each basin in order to account for features that are specific to the
respective geographical region. This can be achieved by considering the four defined domains in the study. To illustrate this approach, Figure 7 depicts the average number of cloud band days per year, computed from 1959 to 2021, overlaid on the Global Precipitation Climatology Project (GPCP) data (Huffman et al., 2020). The GPCP data combines satellite and gauges measurements of precipitation.





The SPCZ (Figure 7a) exhibits a maximum occurrence of cloud bands in the central Pacific, situated at the southeasternmost
mean position of the SPCZ (Vincent et al., 2011). This region, situated south of French Polynesia, experiences approximately
two months of cloud band days per year. This maximum occurrence of cloud bands is positioned between two precipitation
maxima in the central Pacific region: one of these maxima is associated with mid-latitude dynamics, while the other originates
from the SPCZ itself.

In the North Pacific region (Figure 7b), the occurrence of cloud bands displays a peak (35 days of cloud bands per year)
between two areas characterized by heavy precipitation: the BFZ and the Maritime continent. Additionally, in the northeastern
part of the North American continent, the detection of cloud bands suggests a potential association with cloud cover result-
ing from mid-latitude Rossby wave breaking, which instigates convection and can lead to heavy precipitation (Kiladis and
Weickmann, 1992; Knippertz, 2007; de Vries, 2021).

In the southern part of the African continent and the southwestern Indian Ocean (Figure 7c), the distribution of cloud bands
follows a similar pattern to that of the SICZ, which is influenced by the circulation around the Angola Low, the northeastern
monsoon region, and the South Indian Ocean high pressure system that extends over the continent (Cook, 2000; Ninomiya,
2008). This pattern of cloud bands in the region was previously observed by Hart et al. (2012) using OLR data with coarser
resolution (2.5°). Among the four convergence zones, the SICZ displays relatively lower activity in terms of cloud band days
per year, with a maximum of 21 days cloud band days per year. Additionally, this convergence zone demonstrates lower surface
precipitation compared to other convergence zones, mainly owing to the predominance of rainfall during the summer months
across most inland regions and during the winter months along the western Cape coastal areas (Harrison, 1984). Moreover
precipitation is modulated by the Madden–Julian oscillation (Pohl et al., 2007).

Finally, in the case of the SACZ (Figure 7d), the region extending from southern Amazonia to the southeast coastal regions
of Brazil exhibits the highest annual occurrence of cloud bands, with 56 cloud band days per year. Over the South Atlantic
Ocean, lower cloud band occurrences ranging from 21 to 35 days per year are observed. Our findings align with the spatial
distribution of OLR in the presence of an intense and continental SACZ, as defined by Carvalho et al. (2004). Furthermore, the
dominant role of deep convection in the southern Amazon Basin and the southeastward extension of low OLR values into the
Atlantic have been previously highlighted (Liebmann et al., 1999; Carvalho et al., 2002). Precipitation from the GPCP reveals
two distinct regions of strong precipitation. One is located over the southern Amazon Basin, while the other extends over
the South Atlantic Ocean, encompassing parts of south Brazil and Uruguay. The oceanic region is influenced by mid-latitude
dynamics and Rossby wave breaking (Liebmann et al., 1999; Zilli and Hart, 2021).

## 3.3 Climatology and temporal evolution

The use of automated detection techniques facilitates the study of the variability associated with cloud bands. In this regard,
Figure 8 presents the time series of the annual frequency of cloud bands across the four aforementioned domains.
The South Pacific domain stands out with the highest annual occurrence of cloud bands, highlighting the strong convective
activity of the SPCZ (Vincent et al., 2009; Dowdy et al., 2012) deeply entrenched in the tropical region, and including the
Maritime continent, compared to the other three convergence zones. The intrusion of fronts from the mid-latitudes into the





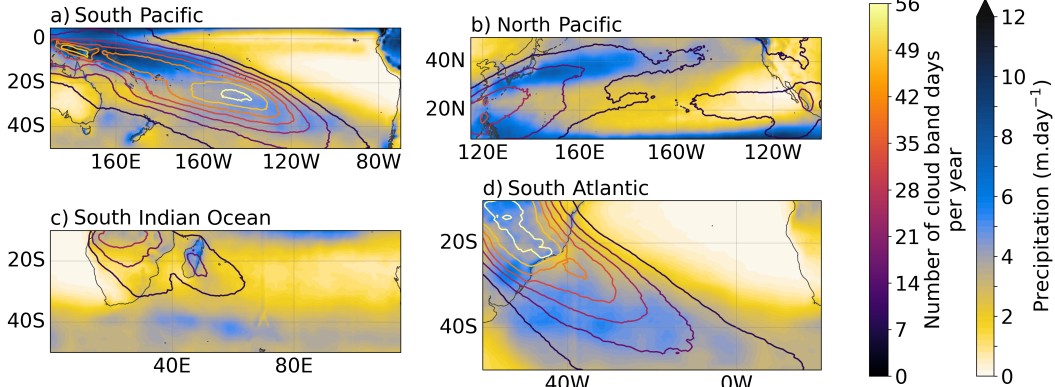

**Figure 7.** Mean precipitation rate in mm.day$^{-1}$ from the GPCP precipitation data from 1983 to 2019 (shading), overlaid by the number of cloud bands per year averaged over the same period (contour interval: 7 cloud band days per year). Contours correspond to contours in Figure 6 but the detection is performed separately for the four domains.

South Pacific further increases the mean annual occurrence of cloud bands, as multiple cloud bands can be present in a single day (Figure 4). The North Pacific domain sees between 100 and 200 cloud band days per year. In this domain, our algorithm

captures cloud bands from the BFZ but also cloud bands associated with eastward-propagating disturbances (Huaman et al., 2020) and atmospheric rivers (Dettinger et al., 2011). Over the South Atlantic domain, the occurrence of cloud bands ranges from 60 to 150. A part of the SACZ and its associated cloudiness is located outside, northwest of the domain we define, which is discussed in section 2.3. Hence, the algorithm cannot detect some of the land-based cloud systems than might account for a part of the cloud band day occurrence. In contrast, the South Indian Ocean (yellow line in Figure 8) demonstrates the

lowest occurrence of cloud bands. The cool and highly variable sea surface temperatures over the SICZ lead to less intense convective activity and cloud band formation (Shannon et al., 1990) compared to other convergence zones. Moreover, the SICZ is a land-based convergence zone whose intensity is partially influenced by surface conditions over southern Africa, with its intensity partially determined by surface conditions over southern Africa. This characteristic results in a stronger annual cycle and intermittent behavior compared to the other convergence zones (Cook, 2000).

The magnitude of these time series is consistent with the spatial patterns of cloud band occurrence displayed in Figure 7.

A similar picture emerges in the seasonal cycle as shown in Figure 9, which represents the annual cycle of cloud bands averaged from 1959 to 2021. The seasonal cycles of the respective domains follow the seasonal cycle of convective activity and precipitation. For example, in the South Pacific, cloud bands are the dominantly present during austral summer, in agreement with findings from Vincent (1994) and Matthews (2012), and the cloud bands shows a strong seasonal cycle with around 3

times more cloud band days per month during austral summer than during austral winter. The variability between seasons in the North Pacific domain is low, with only a twofold increase in cloud band days during the summer season. In the South Atlantic, the magnitude of the annual cycle of cloud bands over the SACZ is similar to the findings of Zilli and Hart (2021). Moreover, because this convergence zone is more strongly rooted in the subtropics and over land, convection and associated



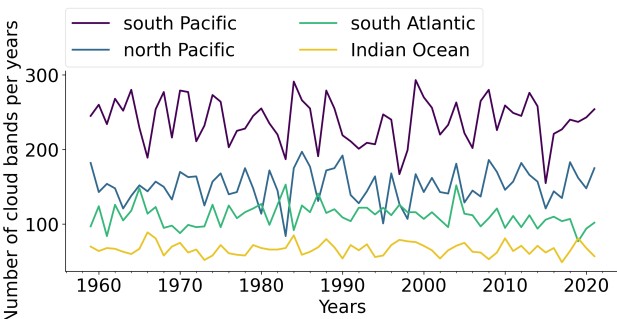

**Figure 8.** Time series of the number of cloud bands per day for each domain defined in Figure 2.

cloudiness are more strongly affected by seasonality (Paegle et al., 2000; Jones and Carvalho, 2002; Carvalho et al., 2004). The

seasonal cycle of cloud band days over the South Atlantic ranges from 2 to 20 cloud bands per month during the winter/dry season and the summer/wet season, respectively. The South Indian Ocean exhibits the fewest cloud bands per month among all the domains. There is an extended period of no or very few cloud bands from June to September. We find more cloud bands per month compared to Hart et al. (2013), who find that the seasonal cycle from 1979 to 1999 ranges from 0 to 5 cloud bands per month, and who used a domain smaller than ours.

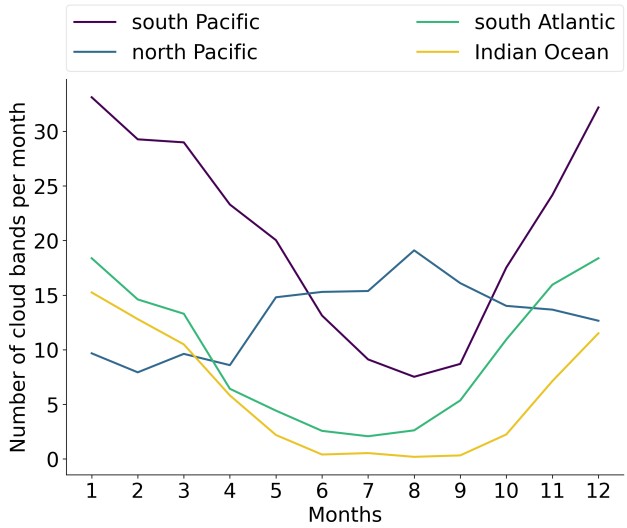

**Figure 9.** Annual cycle of the number of cloud bands per month for each domain defined in Figure 2.



## 4 Discussion and conclusion

In this study we develop a Python-based tool for efficiently detecting atmospheric tropical-extratropical cloud bands, aiming to better understand their spatio-temporal distribution and associated atmospheric processes.

This novel tool combines thresholding and labeling methods, and allows for explicit merging and splitting of cloud bands using conventional area-overlapping. We specifically focus on using OLR data. The identification algorithm is developed to handle a variety of weather and climate datasets, such as high-resolution reanalyses or model simulations. In addition, a flexible interface has been designed so that users can apply their own criteria for the identification of cloud bands or use other variables for detection such as brightness temperature. All user-definable parameters are specified in a configuration file that contains detailed explanations for ease of use. Additional features of interest can be implemented without much coding due to the modular framework design. The package can be run on small or high-performance computers, and can rapidly create cloud band datasets. The algorithm code is publicly available, facilitating further refinement of the method. Various visualization and statistical analysis examples are provided in the package. We demonstrate the capability of this algorithm to detect diagonal cloud bands in different basins chosen based on the location of the major convergence zones.

We also showed the limitations of the employed method. Context and application play a crucial role in selecting threshold values, as different values may be suitable depending on the specific use case. For example, previous studies by Hart et al. (2012) and Hart et al. (2018) primarily examined the mid-latitude section of tropical-extratropical cloud bands, which exhibit less convective activity. These studies adopted a higher OLR threshold, encompassing a wider range of cloud types. We specifically focused on identifying cloud bands characterized by convection, and set a OLR threshold value according to the OLR distribution associated with deep convection.

Alternative thresholding methods were tested, such as global thresholding methods (see appendix A), which automatically select optimal threshold values based on data. While useful for certain applications, if used globally, these techniques may lead to mislabeling features as cloud bands, particularly in situations with significant temporal and spatial variability in OLR values, encompassing regions such as the Tibetan Plateau, South Asian Monsoon, and BFZ. Hence, careful interpretation and domain refinement are important, which necessitates selecting appropriate domain limits. Domains are provided in the code. Using the algorithm in specific domains can therefore improve the accuracy of the detection and allows for a better characterization of cloud bands over convergence zones. Further regional refinements will still be required in order to limit the detection to cloud bands.

Another approach is to use machine learning techniques (Cilli et al., 2020; Beucler et al., 2021; Prabhat et al., 2021), which can automatically select the best threshold values based on the data and the studied phenomena (see Appendix A). These models can evaluate multiple features and their interactions to determine the optimal threshold values for each feature, without requiring manual tuning or manual labeling.

Further developments could include combining cloud features with precipitation features similarly to techniques for tracking MCSs (e.g. Yuan and Houze, 2010; Fiolleau and Roca, 2013; Feng et al., 2022). This approach allows for unrestricted global



tracking and improves the detection of tropical-extratropical cloud bands, while separating cloud cover from non-cloud band regions.

The demonstrated method can then be used for further investigations on the cloud band climatology as well as for studying connections with synoptic processes.

*Code and data availability.*   The open-source software described in this study is made available under the terms and conditions of the BSD3 license. The software can be obtained from GitHub at: https://github.com/romainpilon/cloudbandPy. The exact version of the model used to produce the results used in this paper is archived on Zenodo (https://doi.org/10.5281/zenodo.7989795).

The package includes notebooks and a repository that compiles data for the purpose of enabling new users to easily adopt it for their own research and to ensure reproducibility.

The ERA5 climate reanalysis data (Hersbach et al., 2018, 2020) are publicly available on the Copernicus Climate Change Service (2023) at https://cds.climate.copernicus.eu. The results contain modified Copernicus Climate Change Service information 2020.

The GPCP version 3.2 satellite-gauge combined precipitation data are available at https://disc.gsfc.nasa.gov/datasets/GPCPMON_3.2/
summary.

## Appendix A:  Use of global thresholding method

In image processing applications, thresholding, also known as image segmentation, is a technique used to separate objects or regions of interest from an image based on their pixel intensity values. It involves selecting a fixed threshold value that acts as a cutoff point. The threshold value is a pixel intensity value used to separate the pixels into two groups: those with
intensities above the threshold are assigned to the background, and those with intensities below the threshold are assigned to the foreground. In this study, the foreground corresponds to clouds, the background corresponds to clear sky, and the pixel intensity to the OLR threshold value.

Other and more objective techniques exist, such as global thresholding methods. Contrary to simple thresholding techniques with a fixed threshold, global thresholding methods automatically determine the threshold value by selecting a single value
that minimizes the variance between the foreground and background pixels, based on the histogram of the image. Global thresholding methods hold promise for greater objectivity compared to thresholds determined subjectively by humans, as they do not rely on specific phenomena observed in a limited number of cases.

However, these methods have limitations when the object (the cloud band) of interest in the foreground is not well separated from the background or when values (in this case OLR) are irregular across the image. Moreover, this technique and more
advanced ones, such as adaptive thresholding or edge-based segmentation have been developed mainly for contrasting images with edges, such as for optical character recognition to convert an image of text into text format. Such methods were developed mainly to make details visible throughout data, and may not be applicable to the same degree when studying physical objects without clear boundaries.





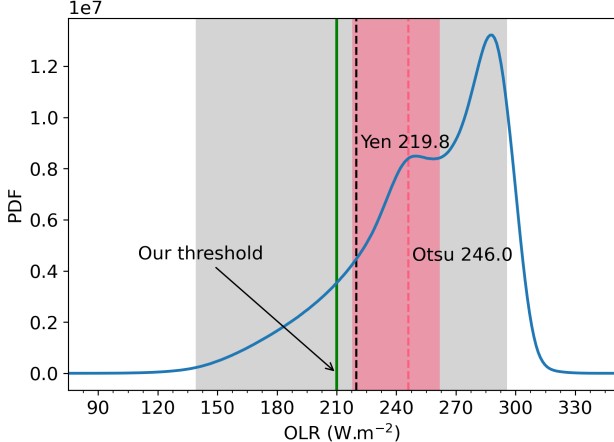

**Figure A1.** Distribution of OLR values for each grid point of the South Pacific domain covering 130°–290°E and 5°N–50°S (see Figure 2). The vertical green line represents the threshold value used in our study. The vertical dashed lines represent the median threshold values obtained from applying the Yen and Otsu automated thresholding methods (see explanations in the text). The gray and pink shaded areas indicate the range of values found for each day of the period, from the minimum to the maximum, for the Yen and Otsu thresholding methods, respectively.

We tested two popular approaches: 1) the Otsu method (Otsu, 1979), which is particularly useful when there is a bimodal

distribution of pixel intensities in the image, such as the OLR distribution shown in Figure A1; 2) the Yen method (Yen et al., 1995), which finds an optimal threshold by minimizing the difference between the probability distributions of pixel intensities between the foreground and background regions at each time step. The Yen method is particularly relevant for identifying features in scattered data.

     Figure A1 shows the probability distribution of OLR over the South Pacific domain, using all grid points and all days from

1960 to 2021. The distribution is bimodal with two peaks at 250 and 295 W.m$^{-2}$. The Yen and Otsu methods yield threshold values of approximately 220 and 246 W.m$^{-2}$ indicated by the vertical lines, respectively. The Otsu median threshold value proves to be excessively high for detecting tropical-extratropical cloud bands composed of convective systems (Massie et al., 2002), hindering the detection of cloud bands composed of warmer clouds with higher OLR values. Consequently, the detected features in Figure A2c and f do not solely represent tropical-extratropical cloud bands.

On the other hand, the Yen median value of around 210 W.m$^{-2}$ aligns more closely with our chosen threshold and has a higher likelihood of accurately detecting cloud bands (Figure A2e). However, in some cases, the value calculated with the Yen method is still too high to effectively detect tropical-extratropical cloud bands, leading our algorithm to merge different features. For example, in Figure A2 (b), the central Pacific cloud band is merged with two mid-latitude troughs. During the development stage of our algorithm, we opted for a more restrictive approach and then expanded the detected feature using

morphological dilation, as described in Section 2.2.





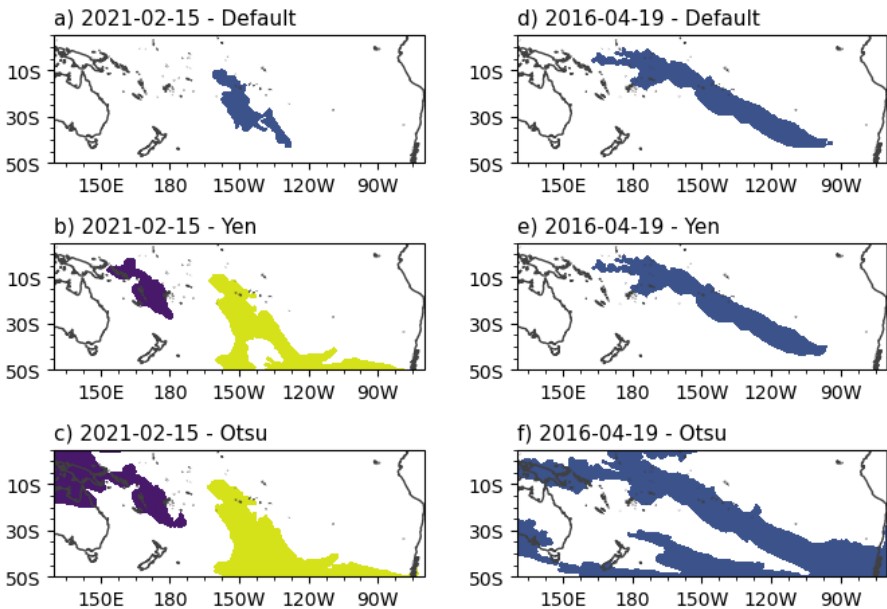

**Figure A2.** Comparison of cloud band detections using various thresholding techniques on 15 February 2021 and 19 April 2016. The images in panels (a) and (d) were processed using the threshold value from this study (i.e. 210 W.m$^{-2}$), while panels (b) and (e) utilized the Yen thresholding technique and (c) and (f) utilized the Otsu thresholding technique.

Moreover, the shaded areas in Figure A2 represent the range of threshold values calculated for each day of the period. The wide range of values indicates a significant daily variability. It is noteworthy that on certain days with no cloud bands, the global thresholding method may still identify a feature to extract from the image, typically characterized by the lowest contiguous OLR values at each time, potentially mislabeling this feature as a cloud band (Figure A2f).

*Author contributions.* R.P. developed the method, made the figures, and wrote the draft manuscript. D.D. contributed to discussions on the method and the writing of the manuscript.

*Competing interests.* The authors declare that they have no competing interests.

*Acknowledgements.* Support from the Swiss National Science Foundation through project PP00P2_198896 to D.D. is gratefully acknowledged.



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
