# Peer review of "cloudbandPy 1.0: An automated algorithm for the detection of tropical-extratropical cloud bands"

_EGUsphere, 2023_

## Author Comment (AC1)

[Figure]

Figure 1. Case study example of the cloud band identification processing steps for 18 April 2016 using OLR over the South Pacific ocean. a) Daily mean OLR data in shading and 210 W.m2 isocontour in yellow, b) binarization, c) morphological dilation, d) connected component labeling with different colors indicating distinct identified features, e) cloud systems with a size below a threshold value are removed and geometrical properties are calculated to filter tropical-extratropical cloud bands, f) cloud band after filtering. The life of each cloud band is tracked between two consecutive time steps (see section 2.4).

[Figure]

Figure 8. Time series of a) the number of cloud bands per year for each domain defined in Figure 2, and b) of the mean cloud band area per year (expressed in km$^2$).

[Figure]

Figure 9. Annual cycle of the number of cloud bands per month (lines) for each domain defined in Figure 2, overlaid by the annual cycle of mean cloud band area (markers: squares - South Pacific, circles - North Pacific, diamonds - South Atlantic, triangles - South Indian Ocean). Marker colors correspond to line colors

---

## Author Response (AR1)

**Reviewer 1**

In this study, the authors developed a new Python-based software to detect cloud bands and show the preliminary results of the detection of tropical-extratropical cloud bands by using this tracking tool. While the research topic is worthwhile, the results are interesting and the logics are easy to follow, there are some room of this study to be improved: For example, the authors should clearly state the advantage as well as the shortness of their tracking software compared with current existed tracking algorithm, to highlight the significance of this study. In addition, the authors should add a schematic flowchart of cloud identification of CloudbandPy v1.0. Moreover, the authors should dig into more detail on the specific properties of each cloud bands they have identified and tracked among four different chosen regions, as well as their changes during 1958~2021, rather than just show roughly how many cloud bands have been identified (more comments please see below). Therefore, I recommend 'Major revision" of the current paper with the following comments.

We appreciate the thoughtful review provided by the reviewer. The comments offer valuable insights that undoubtedly enhance the clarity and depth of our manuscript. In response, we have addressed each point raised and detailed the steps taken to strengthen the paper based on the reviewer's suggestions.

General comments:

1.      Since the manuscript mainly aims to introduce the Python-based tool for efficiently detecting cloud bands, I suggest that the authors should add a schematic flowchart of cloud identification by using CloudbandPy v1.0, to let potential readers easily get the basic flow in L90-L140.

While we believe the manuscript already includes a sufficient number of figures (12 in total), and adding such a figure might overlap with Figure 1. However, recognizing the validity of the reviewer's suggestion, we have replaced Figure 1 with a new figure (see attached document) that combines a schematic flowchart with the cloud band identification processing steps.

2.      What is the potential advantage, as well as the shortness of Cloudband Py v1.0, compared with current existed tracking software, such as FLEXTRKR algorithm (Feng et al. 2021), MOAAP algorithm (Prein et al. 2021), Iterative Rain Cell Tracking (Moseley et al. 2019) and among many others open-source software...? And what is the distinguishing feature of this newly-developed software Cloudband Py v1.0? This is quite important and should be highlighted and discussed in the manuscript, because it will exhibit the scientific significance of this study.

The cloudbandPy software is different from tracking methods mentioned by the reviewer and older, similar mesoscale convective system, and convective cell tracking methods.

Our interest lies in exploring the connections and larger-scale processes and dynamics that lead to the formation of these tropical-extratropical quasi-stationary bands of clouds, often composed of multiple MCSs from the tropics and clouds associated with Rossby-wave breaking (or front intrusion) or enhanced convection generated by the equatorward intrusion of potential vorticity from the extratropics. In some cases, it might resemble what some call atmospheric rivers. These cloud bands are the result of connections that we are studying.

Moreover, the code does not track cloud bands like the mentioned algorithms track features such as MCSs; it does not track the geographical course of the entire cloud bands. Here, the code identifies bands of clouds and "tracks" the life cycle of these features, meaning the tracking only provides information on what the cloud band was at the previous time step and what it will become at the consecutive time step. We have added an additional sentence in Section 2.4 (lines 199-201 in the revised manuscript) to be more specific about the type of tracking used in our study.

However, our software bears a resemblance to the methodology developed by Hart et al. (2012). From a technical standpoint, in comparison to their work, cloudbandPy utilizes a more recent Python version and libraries, making it perceived as more user-friendly. We have adopted an open-source approach for code development.

Additionally, we have extended cloud band detection to tropical regions and initiated a discussion on the limitations inherent in detecting cloud bands (refer to section 2.3). Finally, here in cloudbandPy, we wanted to use the same image segmentation threshold across diverse datasets, prioritizing a physics-centric approach. Our software distinguishes itself by focusing initially on the underlying physics before delving into data processing.

The downsides of our software are as follows:

Firstly, a significant limitation of our software stems from the fact that the detection algorithm uses Outgoing Longwave Radiation (OLR), which is a derived variable dependent on both temperature and emissivity, unlike brightness temperatures for example, which provide a more straightforward and reliable method for cloud detection. OLR is sensitive to variations in surface temperature and atmospheric composition, potentially resulting in altered identifications of cloud features.

Secondly, our software exclusively detects tropical-extratropical cloud bands. While this aligns with the primary objective of the software, it is dependent on the criterion we employ to detect these cloud bands.

We have revised the conclusion to include these considerations (lines 356-357 and lines 364-369) in response to the reviewer's feedback.

Finally, from a technical standpoint, the code does not automatically detect the longitude mapping convention, which may pose challenges in the detection process. Currently, users are required to employ data with the longitude mapping convention used by ECMWF (0 to 360 degrees) or preprocess the data accordingly. Presently, there is no form

of remapping to identify cloud bands that span across basin boundaries defined by the longitudinal constraints of the data.

We also included this point in the conclusion (lines 385).

3.      Why use the ERA5 reanalysis to represent the observed OLR, rather than satellite retrievals? Can the ERA5 represent the observed OLR? What are the uncertainties in ERA5 when compared with the satellite observations? The authors should at least discuss this point and show some sensitivity test on one specific year.

In our study, we opted for ERA5 reanalysis data to represent the observed top-of-atmosphere OLR instead of relying on satellite retrievals for several specific reasons. Our primary objective is to integrate this algorithm into models, leveraging the widespread availability of OLR over brightness temperatures.

To begin with, our study prioritized the fundamental requirements of long-term data, high temporal resolution, and fine horizontal resolution. In contrast to satellite retrievals, which often exhibit limitations such as a comparatively shorter record (the longest being from CLARA data: 1979-present), daily or monthly temporal resolution, or insufficient temporal coverage (as in the case of CERES starting in the 1990s), and horizontal resolution too coarse (NASA GEWEX: 1°), ERA5 provides a continuous and extensive dataset with satisfactory temporal and spatial resolution.

Furthermore, while our current work is based on ERA5 reanalyzes, it's important to note that the developed code is designed to be versatile and adaptable to different datasets. The choice of ERA5 in this context serves as a demonstration, showcasing the capabilities of the method. Researchers have the flexibility to substitute other observational datasets, including satellite retrievals, to suit their specific needs and preferences.

Despite ERA5 generally performing well in representing tropical-mean values of OLR, we acknowledge certain limitations. ERA5's OLR, simulated by the reanalysis system radiation scheme and assimilating satellite radiance data (Hersbach et al., 2020), closely aligns with observed data, but tends to underestimate cloud radiative effects. Noteworthy, distributions of long-wave, short-wave, and total cloud radiative effects at the top of the atmosphere in ERA5 show high consistency with observed values, distinguishing it from other reanalyzes that often exhibit substantial biases (Wright et al., 2020). ERA5 exhibits distributions of long-wave, short-wave, and total cloud radiative effects at the top of the atmosphere that closely align with the observed data.

We have incorporated a new paragraph in section 2.1 of the revised manuscript (lines 89-98) to discuss these points, and we have also added a new reference (Wright et al., 2020) that addresses how reanalyzes compare to satellite retrieval observations.

4.      Please explain the reason why use OLR to identify and track "organized convective cloud systems", rather than Tbb (black body temperature at the cloud top)?

We chose to utilize OLR for identifying and tracking organized convective cloud systems in our algorithm, given its common availability and usability in atmospheric models. Unlike Black Body Temperature at the cloud top (Tbb) or brightness temperatures, which are not typically output directly from models, OLR aligns better with our modeling objectives.

We considered using half-hourly globally merged 4-km pixel-resolution infrared (IR) brightness temperature data (GPM MERGIR), but substantial data gaps (missing values), particularly in extratropical regions, hindered the detection of tropical-extratropical cloud band identification. OLR from reanalyzes, with its consistent availability and reliability, proved more suitable for our purposes.

This aspect has been discussed in the same paragraph as point 3 within section 2.1.

5. Please be more specific on the "calculated properties" when the authors introduce the method (L130), and draw a figure as an example to exhibit how the CloudbandPy v1.0 identify an object, track it and record its various properties (this figure should be the Figure 2, following the current Figure 1).

In the new Figure 1, panel e) (shown in the supplementary document) displays cloud band candidates along with their geometric properties (calculated using the Python library scikit-image): orientation, area, and centroid location. These properties serve two purposes:
- Filtering out tropical-extratropical cloud bands from all detected systems, as outlined in the manuscript (Section 2.2).
- Tracing a connection line between cloud bands that are temporally connected, assumed to be part of the same system, as shown in Figure 1f.

We modified the manuscript (lines 147-151) to be more specific.

6. Since the specific properties of each cloud bands have been calculated and recorded (such as covering area, duration, moving speed...), the authors should provide two or more related figures to exhibit the difference of these properties among four chosen regions (as shown in Figure 9), as well as the changes of these properties during 1958~2021 (as shown in Figure 8).

We appreciate the reviewer's attention to the calculated properties of cloud bands and the tracking. However, it's important to note that in our study, we do not compute duration or moving speed as the analyzed systems are considered quasi-stationary. This choice aligns with the specific objectives of our research, and, as indicated in point 2, our algorithm focuses on a sequence of events to provide insights into the large-scale processes influencing the evolution of cloud bands. For a comprehensive understanding of our approach, we invite the reviewer to refer to Section 2.4, where we elaborate on the details of the inheritance tracking methodology employed in our study.

We appreciate the reviewer's constructive feedback and have made the following revisions to address their comments. In Figure 8 (shown in supplementary document), we have incorporated a new panel (panel b) that displays the yearly mean cloud band area over each domain. Furthermore, in Figure 9, we have overlaid the annual cycle of monthly-averaged cloud band area for each domain. We have also provided brief comments on these new series in the manuscript.

7.      Previous demonstrated that the importance of temporal resolution of source data when someone carry on the object-based tracking (Li et al. 2020), and they suggested that precipitation features can be identified properly by object-based tracking only when dataset with hourly or higher temporal resolution is used. However, the authors converted the hourly ERA5 data into 3hourly ERA5 and carried on the tracking of cloud bands, the authors should comment and discuss about this issue in the revised manuscript. In addition, what is the reason of the threshold of 40% is chosen to detect whether the clouds bands between two consecutive time steps belong to the same track? The reasons that the authors provided in the current manuscript is a little bit too subjective: "...because of the quasi-stationary nature of cloud bands..." (L175-L180).

While we rightly emphasize the significance of utilizing high temporal resolution data (hourly) for tracking precipitation features, it is essential to clarify that such emphasis is beyond the scope of this paper.

Contrary to converting hourly data into 3-hour mean intervals, our algorithm was developed using data at 3-hour intervals, selecting one data point every three hours from the ERA5 data. This decision was made to conserve server space, and it's worth noting that the code is designed to accommodate data at any temporal resolution, whether 1 hour, 12 hours, or daily intervals.

Moreover, we compute the daily mean of the data, which smoothens the cloud features, as explained in lines 111-113 of the manuscript. This process allows us to capture the quasi-stationary characteristics of tropical-extratropical cloud bands. One would need to implement spatial box smoothing for high temporal resolution.

Furthermore, we do not want to track precipitation features, nor do we track them. Instead, our focus is only the detection of quasi-stationary clusters of clouds and MCSs, and gaining an understanding of how large-scale processes/dynamics affect these bands of clouds. We have addressed this comment in the introduction of the revised manuscript (lines 63-66).

Additionally, the decision to set a 40% threshold for the minimum overlap area was made during development of the code. This choice aimed to ensure a meaningful overlap between two cloud bands, preventing the formation of a temporal connection in cases where only a one-pixel overlap occurred. After conducting visual tests (subjective in nature), we observed that overlap values ranging from 0% to 50% yielded similar results for inheritance tracking. Values beyond this range would impact the comprehension of the cloud band life cycle. It's important to note that this tracking does not resemble the

tracking of tropical cyclones or mesoscale convective systems but signifies a "parent-child" relationship between two consecutive cloud bands.
Within the code, users have the flexibility to adjust this threshold value through the configuration file.

In response to the reviewer's comment, the manuscript has been revised (line 204-206), and the code has been updated to incorporate additional comments on this specific topic. The default value of the minimum overlap area has been adjusted to 0, ensuring a temporal connection whenever there is any overlap between cloud bands across two consecutive time steps, unless otherwise specified by the user.

8.      How does the software treat a cloud band when it touches the boundary? This is quite important. For example, in Figure 3, there are two cloud bands (contoured by yellow line and blue line) touch the boundary, if the software still goes on, how could it accurately measure the covering area of these two cloud bands (because some parts of the cloud bands are located outside of the boundary)?

The software operates within a defined domain. With regard to cloud bands touching a boundary, the code implements a solution immediately following the labeling step (as shown in Fig 1d) by merging labeled features that connect through image boundaries, specifically along the longitude axis.

The area of a cloud band does not depend on whether this cloud band crosses the domain's boundaries. In the case of cloud band identified by a blue contour, its area is the total area delimited by the blue contour.
However, the area is only calculated over the domain of detection, and do not extend further outside the domain: 1) the data might be cropped over the domain, or 2) as pointed out in the manuscript (section 2.3), the algorithm might lead to erroneous detection of cloud band.

We have addressed the treatment of cloud bands at the boundary in the revised manuscript. Specifically, a sentence discussing the boundary conditions has been added (lines 139-140) and an additional note about the limitations associated with using a defined domain has been added at the end of section 2.3 (lines 196-197).

Some minor edits:

1.      Why the 10S~10N have been masked out in Figure 6? Did the authors identify and track all the cloud bands over the 50S~50N and then mask out the regions within 10S~10N? Or they just identify and track the cloud bands over 10S~50S and 10N~50N?

We specifically identify cloud bands within the latitudinal range of 10°S to 50°S and 10°N to 50°N, leaving blank areas on the map. This choice was deliberate as we aimed to exclude the ITCZ and extratropical regions where cloud tops are cold and OLR values are often low. We updated the text of section 3.2. A sentence has been added (lines 260-261).

Reference:
1) Feng, Z., Leung, L. R., Liu, N., Wang, J., Houze Jr, R. A., Li, J., ... & Guo, J. (2021). A global high-resolution mesoscale convective system database using satellite-derived cloud tops, surface precipitation, and tracking. Journal of Geophysical Research: Atmospheres, 126(8), e2020JD034202.
2) Prein, A. F., Rasmussen, R. M., Wang, D., & Giangrande, S. E. (2021). Sensitivity of organized convective storms to model grid spacing in current and future climates. Philosophical Transactions of the Royal Society A, 379(2195), 20190546.
3) Moseley, C., Henneberg, O., & Haerter, J. O. (2019). A statistical model for isolated convective precipitation events. Journal of Advances in Modeling Earth Systems, 11(1), 360-375.
4) Li, L., Li, Y., & Li, Z. (2020). Object-based tracking of precipitation systems in western Canada: the importance of temporal resolution of source data. Climate Dynamics, 55(9-10), 2421-2437.

Wright, J. S., Sun, X., Konopka, P., Krüger, K., Legras, B., Molod, A. M., Tegtmeier, S., Zhang, G. J., and Zhao, X.: Differences in tropical high clouds among reanalyses: origins and radiative impacts, Atmos. Chem. Phys., 20, 8989–9030, https://doi.org/10.5194/acp-20-8989-2020, 2020.

Hersbach, H, Bell, B, Berrisford, P, et al. The ERA5 global reanalysis. Q J R Meteorol Soc. 2020; 146: 1999–2049. https://doi.org/10.1002/qj.3803

**Reviewer 2**

Review of " cloudbandPy 1.0: an automated algorithm for the detection of tropical-extratropical cloud bands" by R. Pilon and D. Domeisen.

In this study, the authors introduced a newly developed detection and tracking algorithm specifically designed for tropical-extratropical cloud bands. While they presented preliminary results based on ERA5 reanalysis and discussed potential limitations in their algorithm, the paper lacks a comparative advantage of their method over existing algorithms. Moreover, many parameters in the current algorithm seem arbitrary and require additional validation. Therefore, I recommend 'Major revisions" of the current paper. Please refer to my general comments below:

We appreciate the comments provided by the reviewer. The comments have allowed for improvements of the manuscript and of argumentation.

It is important to note that we addressed not only the limitations in our algorithm but

also those associated with the OLR-based segmentation algorithm for detecting tropical-extratropical cloud bands.

The scope of the paper is not on comparing with different algorithm. Our goal was to develop an open-source, user-friendly tool to address the lack of available options for detecting cloud bands. This tool serves our purpose of connecting large-scale processes that link the tropics to the extra-tropics, with cloud bands acting as a proxy for further studies.

1. The authors should emphasize the novelty and relevance of their work within the research community. Presently, their tool appears similar to existing methods with only variations in thresholds and programming language. In Lines 65-69, the authors claim that "most of the available open-source tools are tailored for the subtropics, and do not treat merging and splitting of cloud bands, and are not optimized to work with different data, and thresholds are dependent on the specific dataset". However, this assertion lacks specific references to support these claims. Several established tracking algorithms—such as the global-scale applicability in Feng et al. (2021) and the consideration of merging and splitting (Huang et al. 2018)—already address these aspects. Furthermore, the authors acknowledge some limitations in their own method but fail to address these mentioned concerns. They should provide clarification on how their method's thresholds remain independent of specific datasets.

Feng, Z., Leung, L. R., Liu, N., Wang, J., Houze Jr, R. A., Li, J., ... & Guo, J. (2021). A global high-resolution mesoscale convective system database using satellite-derived cloud tops, surface precipitation, and tracking. Journal of Geophysical Research: Atmospheres, 126(8), e2020JD034202.

Huang, X., Hu, C., Huang, X., Chu, Y., Tseng, Y. H., Zhang, G. J., & Lin, Y. (2018). A long-term tropical mesoscale convective systems dataset based on a novel objective automatic tracking algorithm. Climate dynamics, 51, 3145-3159.

Thank you for highlighting the lack of clarity in the referenced sentence. In the paragraph in question, when we refer to "tools" we specifically mean cloud band detection algorithms.

We want to emphasize that in our study, our algorithm connects extratropical mesoscale convective systems (MCSs) (fronts and frontal intrusion in the tropics) with tropical MCSs (e.g., cloud SPCZ) into a cloud band. For example, the first reviewer highlighted the MOAAP algorithm (Prein et al. 2021), which separates tropical and extratropical processes, when we hypothesized that some teleconnection might sometimes occur. We acknowledge the suggestion to compare our algorithm with MCSs-detection algorithms, like the one proposed by Feng et al. (2021), Huang et al (2018) or Fiolleau et al (2013), and we recognize the potential benefits of such a comparison, particularly in regions with cloud bands like the South Pacific. However, this comparison goes beyond the scope of our paper, which primarily focuses on cloud band detection.

We have modified the paragraph in question (L69-74) to improve clarity.

This comment parallels the first reviewer's second comment. Our detailed response can be found at (see our response to the first reviewer: https://doi.org/10.5194/egusphere-2023-1184-AC1]).

2. The cloud band identification method appears somewhat arbitrary. Firstly, the authors justify taking the daily mean of the 3-hour OLR dataset to prevent over-segmentation of cloud systems and enhance connectivity.

The algorithm was developed to use different time frequencies (ranging from 1 hour to more). However, for the sake of the amount of data manipulated, we applied a daily mean that keeps but smoothens the signal of cloudiness associated with the cloud bands.
As suggested by the reviewer, we added snapshots of OLR data in Figure 1. We kept the other figures as they were to not clutter the paper with too many figures (already 11 figures, with 2 more panels in the revised version). Furthermore, for Figure 5, snapshots can be seen in Knippertz et al (2005), as indicated in the manuscript.

- However, unlike individual thunderstorms, for cloud bands spread over broad regions with smoothly distributed OLR, using a daily mean seems unnecessary. A snapshot of the 3-hour OLR for cases illustrated in Figs. 1, 3, 4, 5 could provide clarity. The low OLR threshold (210 K) chosen in this study may contribute to over-segmentation.
- The OLR value discussed in Massie et al. (2002) was associated with those overshooting deep convection, which may not be that relevant to the analysis in this study. The authors can test the sensitivity of their results by changing their threshold slightly (within 10-k).
The OLR values discussed in Massie et al. (2002) were associated not only with overshooting convection but also with cirrus clouds and cirrus associated with detraining anvils. We believe this is relevant to our analysis. Additionally, we experimented with different OLR threshold values; for instance, we used 240 $W.m^{-2}$, as suggested by Hart et al. (2012), resulting in detected cloud bands as shown in Fig. A2 (c and f, here with a slightly larger threshold). We also tried 200 $W.m^{-2}$, following the approach in Rosa et al. (2020), which led to a slight over-segmentation. Subsequently, we explored global automatic thresholding techniques to introduce more objectivity, leading to the creation of Figures A1 and A2; however, we remained unconvinced by the results. We have added a sentence stating that the OLR threshold can be adjusted by the user (L125 in the revised manuscript).

- Secondly, the morphological dilation technique lacks clarity in comparison to using a relaxed OLR threshold. This is very arbitrary without considering the real OLR values surrounding the identified cloud band.
We agree with the reviewer that relaxing the OLR values to a larger value would be a better alternative to dilation. We have added this point in the conclusion (L363-369).

- Thirdly, manually determining the size threshold raises concerns regarding its consistency across regions, especially considering the variability mentioned for different regions.

Regarding concerns about the consistency of OLR values across tropical and extratropical regions: Cloudiness from the tropical part of the cloud bands is mostly composed of convective systems (and associated clouds), and cloudiness from the extratropical part of the cloud bands (as mentioned later by the reviewer) are weather systems that can be associated with disturbances (e.g. fronts or cut-off lows). Adapting the algorithm to account for the various processes involved in these cloud bands would be beneficial. We have included this point in the conclusions. However, our primary focus here is to identify when there's a cloud band that extends from the tropics to the extratropics. This is a simple tool to allow us to potentially investigate and understand the larger-scale processes or tropical-to-extratropical dynamics that lead to such cloud events.

- Fourthly, selecting latitude bands (23.5°, 20°, and 27°) to identify cloud bands lacks justification and seems inadequate for moving cloud bands, as evident in Fig. 5.

We agree with the reviewer that the selection of latitudinal range for a minimal cloud band extent is subjective. However, a criterion must be chosen to filter out systems of clouds that do not have a tropical-extratropical extent. We have added a remark in the section 2.3 Cloud band identification (L161-162).

Specific comments:

L31, austral summer usually refers to Dec-Jan-Feb. Vincent (1994) also demonstrated that SPCZ has the largest extension during DJF.
We removed the content of the first parenthesis.

L61, low OLR values can also be caused by cirrus clouds, which are thin and composed of ice crystals
L62, high OLR values are associated with thick low clouds.
For both L61 and L62 (now L62 and L63), we have modified the paragraph to add "cirrus" and the "thick low clouds".

L71, the analysis lacks results from various types of gridded data at different temporal and spatial scales, as mentioned.

While we agree that it would be beneficial, in our study, we opted for ERA5 reanalysis data to represent the observed top-of-atmosphere OLR instead of relying on satellite retrievals for several specific reasons. Our primary objective is to integrate this algorithm into models, leveraging the widespread availability of OLR over brightness temperatures. ERA5 provides long-term data, high temporal resolution, and fine horizontal resolution.

Please refer to our response to the first reviewer, specifically in Response 3, on this matter.

Figure 2, why do the two regions overlap with each other?

As explained in the manuscript, the detection domains are only a suggestion and can be adjusted as desired. Here, the South Atlantic domain tries to cover the SCAZ, and the area covered by cloud bands (yellow shading in Figure 6).
The domain for detecting cloud bands over the SICZ tries to cover a larger area for cases where cloud bands could form in the center of the Indian Ocean, but also for cases where cloud bands could cross the southern part of the African continent.

We conducted tests with other domains (not shown) to evaluate the impact of the easternmost part of the domain covering the South Atlantic Ocean and the westernmost part of the domain covering the South Indian Ocean on the statistics presented in the manuscript. The results indicate that these areas have very little effect on the mean cloud band area per month. This is because only a few parts of cloud bands are detected in these regions during the entire period we cover. However, to optimize the detection of cloud bands, we choose on the domains outlined in this manuscript.

L171, the rationale for adjusting the latitude bands to 10-50°N and 10-50°S isn't clear, as this adjustment may eliminate cloud bands extending across wider latitudinal ranges (e.g., from 10°S to 30°N).

In this section 2.3, we recommend avoiding global cloud band detection to avoid the shortcomings of such an algorithm, which are mentioned in the previous two paragraphs. For example, extending the latitude ranges could lead to false identification of cloud bands, such as over the South American continent, or identify the ITCZ as a cloud band, or even link cloudiness over the Maritime continent with the Baiu frontal zone. We have included a figure (in supplement to this response) showing how Figure 6 would look with such a latitude range.

L178, It would be beneficial to understand if there's a sensitivity analysis conducted regarding the overlapping threshold.

We appreciate you pointing this out. In the revised manuscript, we remove this value and mention a user-defined value.
Additionally, we performed a sensitivity analysis on this overlap threshold and found that overlap values ranging from >0% to 50% yielded similar results for inheritance tracking. 0 (i.e. any positive overlap area is considered) is a bit too much unrestrictive.

We refer to response 7 to the other reviewer's comment for a more detailed response.
We have added a sentence on the conducted test at the same line now L206.

L193, the statement that cloud bands are quasi-permanent and quasi-stationary contradicts the findings presented in Fig. 5. As demonstrated in this well-studied case, cloud bands can move quickly. Cloud bands are often associated with transient weather systems or disturbances that evolve and traverse along the boundaries between tropical and extratropical regions.

We appreciate and agree with the reviewer's observation.

We added Figure 5 to see how well our algorithm can detect a known cloud band (or here a tropical plume) studied by Knippertz (2005), which is indeed a transient weather system similar to those in the Baiu Frontal Zone, the SACZ, or the South Indian Ocean.

The detection workflow was originally designed for the South Pacific, where cloud bands are formed by tropical convective systems (such as those in the SPCZ) and frontal intrusions; the tropical-extratropical cloud bands are then a combination of something quasi-permanent (e.g. cloudiness over the SPCZ) and something transient (the weather systems coming from the mid-latitude dynamics). Furthermore, as shown in Figure 4, it is difficult to talk about the lifetime of a tropical-extratropical cloud band, and tell when a cloud band begins or ends, especially in the case of tropical-extratropical cloud bands associated with the SPCZ (thus deeply rooted in the tropics).
So, we think that it is difficult to talk about lifetime. In the case of the South Pacific, part of the cloud band is quasi-stationary, and in the other basins, the cloud bands are more subtropical.

We have made modifications to the paragraph (L219-225) to enhance precision.

Technical corrections

*1. W. m2 has been used across different sections of the text as the units of OLR.*
Thank you for catching this mistake. We have rectified the units to W.m-2.

*2. The format of references is not consistent throughout the main text.*
We used the LaTeX commands `\citep` and `\citet` consistently throughout the manuscript, using the template provided by Copernicus for its creation. The references in the .bib file were imported from the journal's citation LaTeX exportation tool. We do not see any inconsistency; however, we would be pleased to know which one the reviewer is referring to.

References:

Fiolleau, T. and Roca, R.: An Algorithm for the Detection and Tracking of Tropical Mesoscale Convective Systems Using Infrared Images From Geostationary Satellite, IEEE Transactions on Geoscience and Remote Sensing, 51, 4302–4315, https://doi.org/10.1109/TGRS.2012.2227762, 2013.

Hart, N. C. G., Reason, C. J. C., and Fauchereau, N.: Building a Tropical–Extratropical Cloud Band Metbot, Monthly Weather Review, 140, 4005 – 4016, https://doi.org/10.1175/MWR-D-12-00127.1, 2012.

Knippertz, P.: Tropical–Extratropical Interactions Associated with an Atlantic Tropical Plume and Subtropical Jet Streak, Monthly Weather Review, 133, 2759 – 2776, https://doi.org/https://doi.org/10.1175/MWR2999.1, 2005.

Prein, A. F., Rasmussen, R. M., Wang, D., & Giangrande, S. E. (2021). Sensitivity of organized convective storms to model grid spacing in current and future climates. Philosophical Transactions of the Royal Society A, 379(2195), 20190546.

Rosa, E. B., Pezzi, L. P., Quadro, M. F. L. d., and Brunsell, N.: Automated Detection Algorithm for SACZ, Oceanic SACZ, and Their Climatological Features, Frontiers in Environmental Science, 8, original Research, 2020.